# Determinants of abortion among youth 15–24 in Ethiopia: A multilevel analysis based on EDHS 2016

**Girma Gilano** **\*, Samuel Hailegebreal**

Department of Health Informatics, School of Public Health, Arba Minch University College of Medicine and Health Sciences, Arba Minch, South West Ethiopia

\* gilanog@yahoo.com

## Abstract

### Introduction

Determinants of the magnitude of abortion among women of diverse social and economic status, particularly in Africa poorly understood because of the missing information in most countries. In this study, we addressed abortion and its determinants among youth women of 15–24 ages to provide clear direction for policymaking in Ethiopia.

### Methods

We examined the 2016 Ethiopian demographic health survey data downloaded from the EDHS website after obtaining permission on abortion among 15–24 age women. We applied bivariate and multilevel binary logistic regression. Community and Individual level abortion predictors passed through a three-level binary logistic regression analysis where we used p-value <0.05 and adjusted odds ratios (AOR) with 95% confidence intervals (CI).

### Result

The abortion among the youth population in this study was 2.5%. Factors associated with pregnancy were age group 20–24 2.5(1.6–3.8), youth with one birth 0.65(0.44–0.96), youth with 2–5 births 0.31(0.18–0.55), age ≥18 0.50(0.33–0.76), married 38(17–84), divorced 20 (7–55), birth in the last five years 0.65(0.44–0.96), middle wealth youth 1.7(1.0.4–2.8), being in Amhara0.31(0.11–0.85), and 0.30(0.12–0.77).

### Conclusion

Less abortion occurred in economically poor youths. It is a noble finding; however, the access problem might lead to the result. We observed more abortions in age <18years; those have not given birth until the data collection date. It portrays forth clear policy direction for politicians and all other stakeholders to intervene in the problem. The analysis also showed abortion increased with age. It shows that as age increased, youths disclose abortion which is rare at an early age, and again given an essential clue for the next interventions. The fact in this study is both age and marriage affected abortion similarly. It might be

**Funding:** The author(s) received no specific funding for this work.

**Competing interests:** The authors have declared that no competing interests exist.

because of various culture-related perceptions where it is not appropriate for an unmarried woman to appear with any pregnancy outcome as the reason behind the decreased number of abortions at a younger age. Thus, more attention is required during implementation for unmarried and lower age youth regardless of the magnitude of the abortion.

## Introduction

Abortion is an expulsion of pregnancy tissue and products of conception from the uterus intentional by acts spontaneously [1]. Worldwide, abortion is the proven cause of maternal death [2]. Abortion procedures can be safe or unsafe and studies associated every type with the risk of youth mother and fetus deaths. Previously shreds of evidence showed that pregnancies from youth subjected to frequent termination because of physically and psychologically incomplete growth and development [3]. The problem associated with abortion increase and access is to service is subjective laws and cultural aspects of the communities across the world [4]. Legalizing abortion showed improvement in access and usage but did not reduce the number of estimated abortions [5].

Abortion has different outcomes on the future of youth mothers and the life of the fetus. Other than termination, conceiving during the youth age is also higher risk and correlated with adverse pregnancy outcomes [6]. Some studies associate youth pregnancies with hypertension, eclampsia, premature onset of labor, fetal deaths, premature birth, and pre-eclamptic toxemia delivery outcomes [7].

Individual and community-level factors reported responsible for engagement in abortion and pregnancy terminations among which socio-demographic status, resources, early age marriage, and sexual intercourse dominated [8].

Family background, age, social environment, emotional wellbeing, educational status, employment status, and relationships were the variables reported significantly associated with early age youth pregnancy and abortion early youth pregnancy antaeus, resources, early marriage and sexual intercourse dominated entry. Abortion [9–12]. A study in Gahanna showed education, religious beliefs, health, economic factors, and family factors are responsible. Other factors also determine the decision to undergo abortion [13]. In other studies, knowledge of abortion rights and services increase usage, although quality, accessibility, and safeties of the abortion also affect the decision to pick the service [14]. In Sub-Saharan countries, women who can make reproductive health decisions are more likely to undergo abortion services [15].

Before 2005, the Ethiopian government legalized only the procedure of aborting, in case it required to save the life of a woman [16]. During that time, abortion-related complications were 32% of all maternal deaths [17]. The current law allows abortion in case of rape, incest, and fetal body impairment and if a woman has a physical or mental disability; or if she is younger than 18 years old. It is the policy that semi-liberalizes abortion [17, 18]. The number of abortions increased by a substantial amount among the age group 15-19years following the new law. One in ten youth reported abortion concerning different socio-demographic factors [19]. In 2014, the number of adolescents seeking post-abortion care because of clandestine abortion was higher than among those aborted legally (50% for 18%); between learned and unlearned (14% for 5%). Data in the same year indicated many abortions of all ages increased from 27% to 53% and was 20% for adolescents. And we observed 96,000 induced abortions and 35,000 clandestine abortions [20]. However, most Ethiopians are religious and have a culturally strong identity, that clandestine remained unknown. Abortion with complications may only appear that the reality is very different. [4]. Thus, considering these backgrounds, more

work remained ahead. Studies like this might help to increase information related to these gaps for future policymaking decisions so that adequate youths might get the service. Therefore, determining abortion among the youth and its associated variables expected to be a great asset for this state of affairs in Ethiopia.

## Methods and materials

### Design

We used a cross-sectional study design based on the data from Ethiopian demographic health survey (EDHS) 2016.

### Setting and participants

The Ethiopian population is 112.0 million in 2019 as per the National Bank of Ethiopia and the World Bank. There are nine regions (Tigray, Afar, Amhara, Oromia, Somali, Benishangul, SNNPR (south nation nationalities people's region), Gambela, and Harari) and two city administrations (Addis Ababa and Dire Dawa) in the country. The administration levels went from regions, zones, and through woredas [21]. We downloaded EDHS 2016 dataset for this study purpose and extracted youth females of age 15–24 years consider only those who had complete records and then cleaned and made the data ready for the analysis. In this process, we extracted 6,401 population [22]. EDHS collects data on fertility and childhood mortality levels, fertility preferences, awareness, approval, use of family planning methods, maternal and child health, domestic violence, knowledge, and attitudes toward HIV/AIDS and other sexually among the adult population. The frame of the Population and Housing Census (PHC) contains a complete list including information about the enumeration area (EA) location, type of residence (urban or rural), and the estimated number of residential households which developed for this purpose by the Central Statistical Agency (CSA) used [22]. We accessed the data from the Demographic and Health Survey (DHS) website. It is available at (http://www.dhsprogram.com) requesting registration for permission. The data we got then used only for the research purpose. We kept all data confidential, and we did not identify households or individuals. EDHS approved by the Ethiopian Health Nutrition and Research Institute (EHNRI) Review Board and the National Research Ethics Review Committee (NRERC) at the Ministry of Science and Technology, Ethiopia. As published in the survey report of 2016, they collected verbal informed consent from participants, and the purpose of the study was clear to participants [22]. Participation in the survey was voluntary, and they respected the right to decline.

### Study variables and data management

The outcome variable for this study was abortion. We took it from the EDHS question 'have you ever had a pregnancy termination?' with the response 'yes' if the woman ever had an abortion and otherwise 'no' as a binary outcome. It usually means abortion is any pregnancy outcome of a miscarriage, abortion, or stillbirth [23–25]. The exploratory variables are individual or group variables showing both mother and child, every socio-demographic variable used in the selected EDHS dataset. After downloading the dataset and including it in the study according to the criteria, we cleaned data in Stata v. 15.0. The data then weighted as per sampling weight, primary sampling unit, and strata before analyzing in Stata 15.0.

Finally, we examined abortion in 2016 datasets and discovered the correlation of independent with outcome variables. Individual and group-level predictors of abortion examined using multilevel logistic regression on pooled data from the datasets. Significance level maintained at $p<0.05$ with 95% confidence intervals (CI). Before the multilevel logistic regression

application, we checked all assumptions. Each variable checked on bivariate before introducing into the consecutive multilevel logistic regression models where 0.2 used to include variables to models.

## Results

We pooled 6,401 female populations from EDHS 2016 dataset picked for the current study to examine abortion status in age 15–24. The proportion of abortions became 2.5%. Community-level weighted frequency analyses showed a higher number of participants were from Oromo (36.3%), Amhara (22.5%), and SNNP (20.37%) regions. Three-fourth of the participants were from urban residences. At the individual level, the age group of 15–19 years was 55% of all participants. Those who learned primary education (54.25%); those with no education also accounted for 20%. More than half (56%) of them included in this study were single; 39% were married. Eighty-four percent of youth started sexual intercourse earlier than 18years; 72% had no work during the survey; 48% were in the rich wealth status category; 32% in unfortunate economic status. Among the youth who participated in the study, around 70% have never given birth; (Table 1).

### Multilevel analysis

The null model had 15.7% of the total abortion variations resulted from the variabilities among clusters and the rest from individual differences. The clustering effect shown up here directed us to take multilevel analyses. The median odds ratio of abortion in the null model was 2.1 affirming the differences among the clusters. We cannot be sure if the results of two randomly picked different or similar as we cannot rule out the between cluster variations.

We followed four model building steps to account for the inter-cluster variation stated above. The null model is an intercept-only model that has given the clue of continuing model development. In model one or the individual level, females in the age group 20–24 years were 2.5 times more likely to have an abortion with an AOR of 2.5 (1.6–3.8). Considering birth in the last five years, female who gave at least one birth showed 35% reduced abortion with an AOR of 0.65 (0.44–0.96), and those who gave to 2–5 birth had 69% reduced abortion with an AOR of 0.31 (0.18–0.55) compared to no birth category. Similarly, the odds of being in the abortion group was 45% reduced for age ≥18 years with an AOR of 0.55 (0.36–0.82) than those <18. In contrast, youths who married had 32 times more likely to have an abortion with an AOR of 32(14–70), and those who divorce were 16 times more likely to experience abortion with an AOR of 16.6(6–44) compared to single.

And in model two that is a community level, Amhara, Benishangul, SNNP, and the Gambella had 66%, 71%, & 54% reduced abortion with AOR of 0.34(0.15–0.75), 0.29 (0.11–0.75), 0.46(0.23–0.94), and 0.28 (0.10–77) respectively compared to the Somali region.

In the ultimate or mixed model, females in the age group 20–24 were 2.5 times more likely to have an abortion with an AOR of 2.5(1.6–3.8) relative to age 15–19. The odds of abortion among wealthy youth were 1.7 times more than the odds of poor wealth status females with an AOR of 1.7(1.0.4–2.8). Considering birth in the last five years, females who gave at least one birth had 35% reduced abortion with an AOR of 0.65(0.44–0.96); those who gave 2–5 births had 69% reduced abortion with an AOR of 0.31(0.18–0.55) compared to no birth category.

The odds of being in the abortion group were 50% reduced for age ≥18 years old with an AOR of 0.50(0.33–0.76) compared to age <18. And as in model 1, youths who married had 38 times more likely to report abortion with an AOR of 38(17–84); those who divorced were 20 times more likely to have an abortion with an AOR of 20(7–55) compared to single marital status. And for community level, youth females in the Amhara region were 69% reduced abortion

**Table 1. The descriptive characteristics of the study participants extracted from EDHS-2016 for abortion analysis in 15–24 years aged youths.**

| Variables | Weighted frequency (%) | Variables | Weighted frequency (%) |
|---|---|---|---|
| **Age** | | **Types of residence** | |
| 15–19 | 3380.9(55.04) | Rural | 1,467.33(23.89) |
| 20–24 | 2,761.8(44.96) | Urban | 4,675.4(76.11) |
| **Highest educational level** | | **Religion** | |
| No education | 1,230.3(20.03) | Orthodox | 2,639.53(42.97) |
| Primary | 3,332.60(54.25) | protestant | 1,487.33(24.21) |
| Secondary | 1,184.30(19.28) | Muslim | 1,883(30.65) |
| Higher | 395.55(6.44) | Others | 132.84(2.16) |
| **Marital status** | | **Wealth status** | |
| Single | 3,499(56.97) | Poor | 2,026(32.98) |
| Married | 2,401.7(39.10) | Middle | 1,113.76(18.13) |
| Divorced | 12.78(0.21) | Rich | 3,002.9(48.89) |
| Divorced | 228(3.72) | | |
| **Media exposure** | | **Birth in five years** | |
| No | 3,039.12(49.47) | No birth | 4,338.7 (70.63) |
| Yes | 3,103.62(50.53) | One birth | 1,245.11(20.27) |
| | | 2–5 births | 558.87(9.10) |
| **Ever had birth terminated** | | **Smoke cigarettes** | |
| No | 5,950.2(97.52) | No | 6,116.3(99.57) |
| Yes | 152.6(2.48) | Yes | 26.5(0.43) |
| **Age at first sex** | | **Respondent currently working** | |
| <18 | 5,208.92(84.80) | No | 4,464(72.67) |
| ≥18 | 933.82(15.20) | Yes | 1,678.6(27.33) |
| **Region** | | | |
| Tigray | 497.6 (8.10) | | |
| Afar | 56.2(0.92) | | |
| Amhara | 1,382(22.50) | | |
| Oromia | 2,229(36.29)) | | |
| Somali | 185.8(3.03) | | |
| Benishangul | 66.4(1.08) | | |
| SNNPR | 1,251.4(20.37) | | |
| Gambela | 18.2(0.30) | | |
| Harari | 16(0.26) | | |
| Addis Adaba | 402.9(6.56) | | |
| Dire Dawa | 37(0.60) | | |

with an AOR of 0.31(0.11–0.85); those living in Benishangul had 70% a reduced abortion with an AOR of 0.30(0.12–0.77); (Table 2).

In Table 3 below, we showed the comparison of each effect of model 0–3. We saw decreased variance from ICC, media odds ratio, and deviance. The log-likelihood ratio and proportional change in variances increased as global expectation. Specifically, the lower deviance shows good model fitness.

## Discussion

Our study examined abortion both at the individual and community levels. For this purpose, we pooled 6,401 samples of age 15-24years female youth population from the Ethiopian

**Table 2. Multilevel logistic regression on individual and community-level factors associated with youth abortion.**

| Variables | Model 0 | Model I | Model II | Model III |
|---|---|---|---|---|
| **Age** | | | | |
| 15–19 | - | 1 | | |
| 20–24 | - | 2.5(1.6–3.8)*** | | 2.5(1.6–3.8)*** |
| **Wealth status** | | | | |
| Poor | - | 1 | | |
| Middle | - | 1.5(0.92–2.44) | | 1.7(1.0.4–2.8)** |
| Rich | - | 1.04(0.65–1.62) | | 0.87(0.49–1.5) |
| **Birth in five years** | | | | |
| No births | - | 1 | | |
| One birth | - | 0.65(0.44–0.96)** | | 0.65(0.44–96)** |
| 2–5 births | - | 0.31(0.18–0.55)*** | | 0.31(0.17–0.53)*** |
| **Age at the first sex** | | | | |
| <18years | - | 1 | | |
| ≥18years | - | 0.55(0.36–0.82)* | | 0.50(0.33–0.76)* |
| **Marital status** | | | | |
| Single | - | 1 | | |
| Married | - | 32(14–70)*** | | 38(17–84)*** |
| Widowed | - | Empty | | Empty |
| Divorced | - | 16.6(6–44)*** | | 20(7–55)*** |
| Region | | | | |
| Tigray | - | | 0.68(0.36–1.3) | 0.75(0.32–1.7) |
| Afar | - | | 1.2(0.67–2.3) | 0.88(0.46–1.6) |
| Amhara | - | | 0.34(0.15–0.75)* | 0.30(0.12–0.77)** |
| Oromia | - | | 0.66(0.35–1.2) | 0.72(0.35–1.4) |
| Somali | - | | 1 | |
| Benishangul | - | | 0.29(0.11–0.75)* | 0.31(0.11–0.85)** |
| SNNPR | - | | 0.46(0.23–0.94)** | 0.95(0.39–2.3) |
| Gambela | - | | 0.28(0.10–77)** | 0.32(0.10–1.03) |
| Harari | - | | 1.03(0.49–2.14) | 0.82(0.37–1.7) |
| Addis Adaba | - | | 0.63(0.28–1.38) | 1.3(0.55–3.3) |
| Dire Dawa | - | | 0.41(0.16–1.03) | 0.39(0.15–1.03) |

**NB**:

* = p<0.01

** = p<0.05, &

*** = p<0.001.

**Table 3. Model comparison and random effect distribution on determinants of abortion among youths in Ethiopia.**

| Random effect model comparison | Model 0 | Model 1 | Model 2 | Model 3 |
|---|---|---|---|---|
| Variance | 0.61 | 0.48 | 0.40 | 0.32 |
| Inter-cluster correlation(ICC) | 0.157 | 0.13 | 0.10 | 0.088 |
| Log likelihood ratio(LLR) | -780 | -645 | -764 | -632 |
| Deviance | 1560 | 1290 | 1528 | 1264 |
| Proportional change in variance(PCV) | Ref | 0.21 | 0.34 | 0.47 |
| Media odds ratio (MOR) | 2.1 | 1.9 | 1.8 | 1.7 |

demographic health surveillance dataset of 2016. Participants of this study are within the age group of 15–24 years old; however, 40% of them were in marriage and exposed to different youth pregnancy-related problems like abortions at most. The analysis showed, 2.5% of them were at least aborted once. The magnitude is incongruent with the studies conducted in Ethiopia and Nigeria [23, 26, 27]. The reason might be the socio-cultural influences where some youth might follow clandestine abortion in some areas. Economically, 33% of youth stayed poor; 73% were workless during the survey. It is consistent with a study by Van Rensburg that showed societal poverty, unemployment, and other socio-demographic factors affected youth pregnancies and might lead to abortion [28]; (Table 1).

Abortion was increased with increased age so that age group 20–24 had more abortion than15-19 The finding is in agreement with that of Nigerian DHS where abortion was 2.34 higher among those aged 20–24 than youths aged 15–19 [23]. It might show that age is the risk of abortion during the younger time because of the lack of information and less self-relying decision making.

We found middle wealthier youths experienced more abortions compared to the poor ones. Other Studies suggested that abortion has been affected by economic status. It was not in some analyses of the same dataset [19–21]. It means youths with financial capacity could abort in case of induction than those who are poor. Youth mothers who gave at least one or more births in the last five years had more abortion history. Studies in Mozambique and Ghana agree with this finding [29]. The reason might be the increased frequency of pregnancy increased the risk of abortions in married youths. Much of the association was among the age group of greater than 18 years [22, 23, 30]. These might be due to the increase in knowledge and self-responsibilities when youth are more mature. Similarly, both the married and the divorced youth mothers had an increased risk of abortion relative to the unmarried ones. The finding is consistent with other studies in Ethiopia, another Africa, and non-African countries [22, 23, 31]; (Table 2).

The fact that divorce youth had an increased abortion might indicate freedom of decision-making on reproductive matters. We compared other regions to Somali so that Amhara and Gambella showed less abortion correlation. From 2000–2016 EDHS pregnancy termination was lower in these regions(Amhara & Gambella) [19]. The event was not so clear whether it is due to actual number or cultural restriction more strong in these regions. However, the information obtained from this study might be additional paramount for further investigation and policymaking. EDHS follow a kind of data collection and methodologies which are internationally standard and scientifically applied in every country. The clustering effect might be another issue during sampling. Furthermore, the economic status classification was country-specific. However, to account for the limitations, we applied weighting technique and Segregated or multilevel analysis to handle variations among clusters; (Table 3).

## Conclusion

We provided information about community and individual/ contextual level analyses of abortion among women aged 15–24 years in Ethiopia. The findings were relatively lower than other African countries like Nigeria, Uganda, and South Africa; but, it is necessary to remember that we used weighted data for analysis and reveals a much more country-level profile. At an individual level, we recognized association with age, wealth index, marital status, birth in the last five years, and age at first sexual intercourse. Amhara and Benishangul region have shown an association with youth abortion. The analysis also showed abortion increased with age. As age advances, youth disclose abortion increases that an essential clue for the next interventions related to youth economic status. Various cultures influenced perceptions and

inability to make reproductive decisions and need more attention behind the decreased abortion of lower age and unmarried youth. The authors look forward to large-scale studies that consider abortion disaggregation within primary data to account for the current limitations.

## Supporting information

**S1 File.**
(DTA)

## Acknowledgments

The authors are very thankful for the responsible DHS for they have given permission to use the data and all other stakeholders involved directly or indirectly.

## Author Contributions

**Conceptualization:** Girma Gilano, Samuel Hailegebreal.

**Formal analysis:** Girma Gilano, Samuel Hailegebreal.

**Methodology:** Girma Gilano, Samuel Hailegebreal.

**Resources:** Girma Gilano.

**Writing – original draft:** Girma Gilano.

**Writing – review & editing:** Samuel Hailegebreal.

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
