## [Decision Letter · Decision Letter 0]

25 Jan 2021

PONE-D-20-40278

Pregnancy termination and determining factors among youth 15-24 in Ethiopia: A

multilevel analysis based on EDHS 2016

PLOS ONE

Dear Dr. Kaso,

Thank you for submitting your manuscript to PLOS ONE. After careful consideration, we feel that it has merit but does not fully meet PLOS ONE’s publication criteria as it currently stands. Therefore, we invite you to submit a revised version of the manuscript that addresses the points raised during the review process.

Two experts in the field handled your manuscript, and we are appreciative of their time and contributions. Although some interest was found in your study, several major concerns arose that require your attention. Please address ALL of the reviewers' comments in your revised manuscript and response-to-reviewers document.

We look forward to receiving your revised manuscript.

Kind regards,

Frank T. Spradley

Academic Editor

PLOS ONE

2.) We suggest you thoroughly copyedit your manuscript for language usage, spelling, and grammar. If you do not know anyone who can help you do this, you may wish to consider employing a professional scientific editing service.  

3.) Please amend either the abstract on the online submission form (via Edit Submission) or the abstract in the manuscript so that they are identical.

4.) Your ethics statement should only appear in the Methods section of your manuscript. If your ethics statement is written in any section besides the Methods, please move it to the Methods section and delete it from any other section. Please ensure that your ethics statement is included in your manuscript, as the ethics statement entered into the online submission form will not be published alongside your manuscript.

Reviewers' comments:

Reviewer's Responses to Questions

**Comments to the Author**

1. Is the manuscript technically sound, and do the data support the conclusions?

Reviewer #1: Partly

Reviewer #2: No

2. Has the statistical analysis been performed appropriately and rigorously? 

Reviewer #1: Yes

Reviewer #2: I Don't Know

3. Have the authors made all data underlying the findings in their manuscript fully available?

Reviewer #1: No

Reviewer #2: Yes

4. Is the manuscript presented in an intelligible fashion and written in standard English?

Reviewer #1: Yes

Reviewer #2: No

5. Review Comments to the Author

Reviewer #1: The authors were attempting, using multivariate binary logistic regression, to identify the determinants/factors behind pregnancy termination while holding other variables constant. The topic is new, specific, and has a valuable contribution to the existing literature and different stakeholders in the country. The paper is organized logically and structured clearly. The purpose and significance of the study are clearly stated, and the research method is appropriate. The method applied in the study to analyze the research topic is praise-worthy. There is a good consistency of analysis throughout the whole paper. Besides, the results are presented explicitly by the tables and figures. However, there are still some parts that need to be modified in the paper. The author is suggested to make some revisions so that the paper will be better.

Suggestions

It is suggested to change the title to ‘determinants of pregnancy termination among youth 15-24 in Ethiopia’ because the paper did not talk about pregnancy termination except the prevalence of it

The introduction lacks sound justification to undertake the study, for example, the severity of the problem, impacts of the problem if not studied

The discussion part did not include the limitation of the study, the study has serious limitations, for example, disproportionate sample size among regions, unjustified age, and socioeconomic status category, which statically changed the results of the study.

There is a serious grammar problem majorly in the methodology and result part

Reviewer #2: At present, this article needs major restructuring to aid clarity, offer analyses and contribute to the literature. Some of these- like situating the study within existing literature- are easier to deal with than others (please see expanded comments on this below). In its present form, I do not recommend publication.

One of the main issues for me, however, is the lack of clarity on the variables used and the analyses of these variables (and their subsequent implications). On pg. 4, the authors write that 'Pregnancy termination is usually to mean any pregnancy outcome of a miscarriage, abortion, or stillbirth'. While some DHS data are collected in this way, without disaggregating between the three (conceptually, empirically, medically, and policy-wise) very different concepts, I'm unclear on how this has been treated within the text. My understanding here is that data used in this study are not specific to (induced) abortion, but include miscarriages and stillbirth (it is possible to disaggregate, as evidenced by the same 2016 EDHS data used in this article https://doi.org/10.1371/journal.pone.0235382). The implications of this are immense as the study suggests that (i) miscarriages, stillbirths and induced abortion are equivalent (which skews the results anyway), and (ii) these are voluntary terminations/are decisions made by young women - when, of course, miscarriages and stillbirths are not deliberate decisions. The authors suggest that their analyses on 'pregnancy termination' can tell us about young peoples' secrecy- this is not evidenced by the data alone, nor is it situated in the literature so it is hard to follow this leap. Just as it is to follow through to the suggestions of this being relevant for policymakers- how so? The authors could do more to expand on this and with specific examples, but once again with the caveat of what it is they are analysing and the limits of that analysis.

The other large issue for me in this article is the assumption underpinning the analyses. The authors do not set this out (despite alluding to it in their methods section). The framings of pregnancy termination- tied to age and marriage, for example (see pg 10, 'For more surprise, when youths married unwanted or unintended pregnancies might not go away') could do with being located within existing knowledge on termination, unintended pregnancies etc. The link, based on these framings and data analyses, with safety of pregnancy termination (pg. 10, 'unsafe pregnancy termination attributes immensely [...]' is not evidenced. It also individualises the 'youth' responses- in the conclusion again, where the authors suggest that single 'youth' hide everything about pregnancy termination, rather than the contexts that may make it difficult/dangerous for them to divulge. The article also does not contend with the range of factors and issues that surround pregnancy termination.

To restructure this article to bring it up to the level of publication, there are major reviews required. These include:

(1) Disaggregating the pregnancy termination variable to focus specifically on pregnancy termination (i.e. induced abortion). This will probably require analysing the data again.

(2) Situating the analyses within the literature in a more nuanced and coherent way- on pregnancy termination (e.g., the authors cite in the introduction, a paper by Ganatra et al, that reports data on abortion according to a spectrum of safety, but the authors report this within a binary- which, once again, raises questions about how the analyses are presented later on), on pregnancy termination in Ethiopia (for example, see literature published in PLOS One: https://doi.org/10.1371/journal.pone.0235382) and Sub-Saharan Africa ( https://doi.org/10.1371/journal.pone.0235329), adolescents/young people and pregnancy termination in Ethiopia (https://www.guttmacher.org/news-release/2018/first-study-incidence-abortion-among-ethiopian-adolescents-released, https://www.guttmacher.org/fact-sheet/adolescent-abortion-ethiopia, https://doi.org/10.1186/s12889-019-6845-7, https://doi.org/10.1016/j.jadohealth.2017.12.015).

(3) We also need to understand the study context better: what is the law in Ethiopia on pregnancy termination, how does it treat young peoples' access to abortion? What are the social barriers?

(4) Language: there is a lot of 'youth' or 'teen' used in this article- even where it is not applicable or appropriate (e.g., ages 15-24 .. what is understood by 'teen' here?). Youth as a 'catch all' term obfuscates- especially if the data are specific to young women and girls' experiences? If so, that needs to be accounted for clearly in the text- and the gendered implications of this.

6. PLOS authors have the option to publish the peer review history of their article (what does this mean?). If published, this will include your full peer review and any attached files.

Reviewer #1: **Yes: **Zewdu Girma Shifaw

Reviewer #2: No

---

## [Author Response · Author response to Decision Letter 0]

10 Feb 2021

Firstly, we are grateful to esteemed reviewers, the board of editors, and all involved stakeholders for making this work eligible for publication in PLOS ONE journal. 

In this section, we tried to respond to each concern raised by the esteemed reviewers. We provided the response in the similar sequence they appeared in the feedback form. Some of the points might be raised differently by reviewers and the response also considered them accordingly. 

1. Reviewer #1

It is suggested to change the title to ‘determinants of pregnancy termination among youth 15-24 in Ethiopia’ because the paper did not talk about pregnancy termination except the prevalence of it

Determinants of abortion among youth 15-24 in Ethiopia: A multilevel analysis based on EDHS 2016 

The introduction lacks sound justification to undertake the study, for example, the severity of the problem, impacts of the problem if not studied

Before 2005, the Ethiopian government legalized only the procedure of abortion, in case it is required to save the life of a woman(16). During that time, abortion-related complications were 32% of all maternal deaths(17). The current law allows abortion in case of rape, incest, and fetal impairment; if a woman has physical or mental disabilities; or if she is younger than 18 years old. It is the policy that semi liberalizes abortion(17)(18). After this, the number of abortions increased by some amount especially among the age group of 15-19. From other pieces of evidence, one in ten youth reported abortion concerning different socio-demographic factors(19). In 2014, the number of adolescents seeking post-abortion care due to clandestine abortion was greater than those aborted legally (50% vs 18%); between learned and unlearned (14% vs 5%). Data in the same year showed, all types of abortion of all ages increased from 27% to 53%, and was 20% for adolescents. And we observed 96,000 induced abortions and 35,000 clandestine abortions(20). However, the fact that most Ethiopians are religious and have culturally strong identities, we might saw clandestine abortion with complications, and the real could be very large(4). Thus, considering these backgrounds, more work remained ahead and studies like this might be helpful to increase information related to these gaps for future policymaking decisions so that adequate youths might obtain the service. Therefore, determining abortion among youths and its associated variables expected to be a great asset for this state of affairs in Ethiopia

The discussion part did not include the limitation of the study, the study has serious limitations, for example, disproportionate sample size among regions, unjustified age, and socioeconomic status category, which statically changed the results of the study

EDHS follow a kind of data collection and methodologies which are internationally standard and scientifically applied in every country so that most limitation might be tolerated. The clustering effect might be another issue during sampling. Furthermore, the economic status classification was dependent on the country’s economic status. However, to account for the limitations, we applied weighting techniques, and Segregated or multilevel analysis to handle variations among clusters

There is a serious grammar problem majorly in the methodology and result part

Methods and materials

Design 

A cross-sectional study design deployed in Ethiopia based on the data from Ethiopian demographic health surveillance (EDHS) 2016. 

Setting and participants 

 World Bank and the national bank of Ethiopia estimated the total population of Ethiopia at 112.0 million people in 2019. The country is divided into 9 regions (Tigray, Afar, Amhara, Oromia, Somali, Benishangul, SNNPR(south nation nationalities people’s region), Gambela, and Harari) and 2 city administrations (Addis Ababa and Dire Dawa). The administration levels went from regions, zones, and through woredas

(21). We downloaded EDHS 2016 dataset for this study purpose and extracted youth females of age 15-24 considering only those who had complete records and then cleaned and made the data ready for the analysis. In this process, we extracted 6,401 samples (22). EDHS collects data on fertility and childhood mortality levels, fertility preferences, awareness, approval, use of family planning methods, maternal and child health, domestic violence, knowledge, and attitudes toward HIV/AIDS and other sexually among the adult population. The frame of the Population and Housing Census (PHC) contains a complete list including information about the enumeration area (EA) location, type of residence (urban or rural) and the estimated number of residential households which developed for this purpose by the Central Statistical Agency (CSA) used(22). We accessed the data from the Demographic and Health Survey (DHS) website (http://www.dhsprogram.com) which request registration for permission. The data we got then used only for the research purpose. We kept all data confidential, and no effort was made to identify households or individuals. EDHS approved by the Ethiopian Health Nutrition and Research Institute (EHNRI) Review Board and the National Research Ethics Review Committee (NRERC) at the Ministry of Science and Technology, Ethiopia. As published in the survey report of 2016, they collected verbal informed consent from participants, and the purpose of the study was clear for participants (22). Participation in the survey was voluntary and they respected the right to decline.

Study variables and data management

The outcome variable for this study was ‘abortion’ extracted from an EDHS question assessed through ‘have you ever had a termination of pregnancy’ with the responses ‘yes’ if the women ever had an abortion and otherwise ‘no’ as a binary outcome. Abortion is usually mean any pregnancy outcome of a miscarriage, abortion, or stillbirth;(23) (24) (25). Considering all the limitations of the secondary data as they are, for this study purpose, the term abortion was appropriate as an all-inclusive indicator. The exploratory variables are individual or group variables showing both mother and child, every socio-demographic variable used in the selected EDHS dataset. After downloading the dataset and including it in the study according to the criteria, we cleaned data in Stata v. 15.0. The data then weighted as per sampling weight, primary sampling unit, and strata before analyzing in Stata 15.0 

Finally, we examined abortion in 2016 datasets and discovered the correlation of independent with outcome variables. Individual and group-level predictors of abortion examined using multilevel logistic regression on pooled data from the datasets. Significance level maintained at p<0.05 with 95% confidence intervals (CI). Before the multilevel logistic regression application, we checked all assumptions. Each variable checked on bivariate before introducing into the consecutive multilevel logistic regression models where 0.2 was used to include variables to models. 

Results

We pooled 6,401 youth female population from EDHS 2016 dataset picked for the current study to examine abortion status in age 15-24. The proportion of abortions became 2.5%. Community-level weighted frequency analyses showed the larger number of participants were from Oromo (36.3%), Amhara (22.5%), and SNNP (20.37%) regions. Three-fourth of the participants were from urban residences. At the individual level, the age group of 15-19 was accounted for 55% of all participants. More than half of the participants were only learned primary education (54.25%) and no education also accounted for 20%. More than half (56%) of youths included in this study were single and 39% were married. Eighty-four percent of youths started sexual intercourse earlier than 18years and 72% of them had no work during the survey; however, 48% of them were in the rich wealth status category and 32% in poor. The birth was not experienced in70% of youths and half of them had no media exposure

Multilevel analysis 

The null model had 15.7% of the total abortion variations coming from the variability among clusters and the rest from individual differences. The clustering effect shown up here more directed us to take multilevel analysis. The median odds ratio of abortion in the null model was 2.1 affirming the differences among the clusters. This means we cannot be sure if the results of two randomly picked clusters different or similar as we cannot rule out the between cluster variations.

We followed four model building steps to account for the inter-cluster variation stated above. The null model is an intercept only model but, has given the clue of continuing model development. In model I which is the individual level, females in the age group 20-24 were 2.5 times more likely to carry out abortion with an AOR of 2.5 (1.6-3.8). Considering birth in the last five years, female with at least one birth showed 35% reduced abortion with an AOR of 0.65 (0.44-0.96), and those who gave to 2-5 births had 69% reduced abortion with an AOR of 0.31 (0.18-0.55) compared to no birth category. Similarly, the odds of being in the abortion group was 45% reduced for age ≥18 years with an AOR of 0.55 (0.36-0.82) than those <18. In contrast to this, youths who married had 32 times more likely to carry out abortion with an AOR of 32(14-70), and those who made divorce were 16 times more likely to experience abortion with an AOR of 16.6(6-44) compared to single.

And in model 2 which is a community level, Amhara, Benishangul, SNNP, and the Gambella had 66%, 71%, & 54% reduced abortion with AOR of 0.34(0.15-0.75), 0.29 (0.11-0.75), 0.46(0.23-0.94), and 0.28 (0.10-77) respectively compared to the Somali region.

In the ultimate or mixed model, females in the age group 20-24 were 2.5 times more likely to carry out abortion with an AOR of 2.5(1.6-3.8) relative to age 15-19. The odds of abortion among females with good wealth status were 1.7 times more than the odds of poor wealth status females with an AOR of 1.7(1.0.4-2.8). Considering birth in the last five years, females with at least one birth had 35% reduced abortion with an AOR of 0.65(0.44-0.96) and those who gave 2-5 births had 69% reduced abortion with an AOR of 0.31(0.18-0.55) compared to no birth category. 

The odds of being in the abortion group were 50% reduced for age ≥18 years old with an AOR of 0.50(0.33-0.76) compared to age <18. And as in model 1, youths who married had 38 times more likely to report abortion with an AOR of 38(17-84) and those who divorced were 20 times more likely to have an abortion with an AOR of 20(7-55) compared to single marital status. And for community level, youth females in the Amhara region were 69% reduced abortion with an AOR of 0.31(0.11-0.85) and those living in Benishangul had 70% reduced abortion with an AOR of 0.30(0.12-0.77)

2. Reviewer #2

(1) Disaggregating the pregnancy termination variable to focus specifically on pregnancy termination (i.e. induced abortion). This will probably require analyzing the data again.

Determinants of abortion among youth 15-24 in Ethiopia: A multilevel analysis based on EDHS 2016

The outcome variable for this study was ‘abortion’ extracted from an EDHS question assessed through ‘have you ever had a termination of pregnancy’ with the responses ‘yes’ if the women ever had an abortion and otherwise ‘no’ as a binary outcome. Abortion is usually mean any pregnancy outcome of a miscarriage, abortion, or stillbirth;(23) (24) (25). Considering all the limitations of the secondary data as they are, for this study purpose, the term abortion was appropriate as an all-inclusive indicator. The exploratory variables are individual or group variables showing both mother and child, every socio-demographic variable used in the selected EDHS dataset. After downloading the dataset and including it in the study according to the criteria, we cleaned data in Stata v. 15.0. The data then weighted as per sampling weight, primary sampling unit, and strata before analyzing in Stata 15.0 

(2) Situating the analyses within the literature in a more nuanced and coherent way- on pregnancy termination (e.g., the authors cite in the introduction, a paper by Ganatra et al, that reports data on abortion according to a spectrum of safety, but the authors report this within a binary- which, once again, raises questions about how the analyses are presented later on), on pregnancy termination in Ethiopia (for example, see literature published in PLOS One: https://doi.org/10.1371/journal.pone.0235382) and Sub-Saharan Africa ( https://doi.org/10.1371/journal.pone.0235329), adolescents/young people and pregnancy termination in Ethiopia (https://www.guttmacher.org/news-release/2018/first-study-incidence-abortion-among-ethiopian-adolescents-released, https://www.guttmacher.org/fact-sheet/adolescent-abortion-ethiopia, https://doi.org/10.1186/s12889-019-6845-7, https://doi.org/10.1016/j.jadohealth.2017.12.015).

Abortion is an expulsion of pregnancy tissue and products of conception either intentional by acts or spontaneous(1). Worldwide abortion is the proven cause of maternal death(2). Abortion procedures can be safe or unsafe but studies associated every type of it with the risk of youth mother and fetus deaths. Previous shreds of evidence showed that pregnancies from youth subjected to frequent termination because of physically and psychologically incomplete growth and development(3). The problem resulting from abortion increase as access has been also influenced by laws and cultural aspects of the communities across the world(4). Legalizing abortion showed improvement in access and usage but do not reduce the number of estimated abortions(5).

 Abortion has different outcomes on the future of youth mothers and the life of the fetus. Other than termination, conceiving during the youth age by itself is a higher risk and correlated with adverse pregnancy outcomes (6). Studies highlighted that youth pregnancies are additionally associated with hypertension, eclampsia, premature onset of labor, fetal deaths, premature, and pre-eclamptic toxemia delivery outcomes(7). 

Individual and community-level factors are responsible for engagement in abortion and pregnancy terminations among which socio-demographic status, resources, early marriage, and sexual intercourse dominated(8)

. Family background, age, social environment, emotional wellbeing, educational status, employment status, and relationships were the variables significantly associated with early youth pregnancy and abortion(9) (10), (11), (12). A study in Gahanna showed education, religious beliefs, health, economic factors, and family made a maximum impact where other factors determined the decision to undergo abortion (13). Another study indicated good knowledge of abortion rights and services increase usage, even so, quality, accessibility, and safeties of the abortion were also assumed to affect the decision to pick the service(14). In Sub-Saharan countries, women who can make reproductive health decisions are more likely to undergo abortion services(15). 

Before 2005, the Ethiopian government legalized only the procedure of abortion, in case it is required to save the life of a woman(16). During that time, abortion-related complications were 32% of all maternal deaths(17). The current law allows abortion in case of rape, incest, and fetal impairment; if a woman has physical or mental disabilities; or if she is younger than 18 years old. It is the policy that semi liberalizes abortion(17)(18). After this, the number of abortions increased by some amount especially among the age group of 15-19. From other pieces of evidence, one in ten youth reported abortion concerning different socio-demographic factors(19). In 2014, the number of adolescents seeking post-abortion care due to clandestine abortion was greater than those aborted legally (50% vs 18%); between learned and unlearned (14% vs 5%). Data in the same year showed, all types of abortion of all ages increased from 27% to 53%, and was 20% for adolescents. And we observed 96,000 induced abortions and 35,000 clandestine abortions(20). However, the fact that most Ethiopians are religious and have culturally strong identities, we might saw clandestine abortion with complications, and the real could be very large(4). Thus, considering these backgrounds, more work remained ahead and studies like this might be helpful to increase information related to these gaps for future policymaking decisions so that adequate youths might obtain the service. Therefore, determining abortion among youths and its associated variables expected to be a great asset for this state of affairs in Ethiopia

(3) We also need to understand the study context better: what is the law in Ethiopia on pregnancy termination, how does it treat young peoples' access to abortion? What are the social barriers?

Before 2005, the Ethiopian government legalized only the procedure of abortion, in case it is required to save the life of a woman(16). During that time, abortion-related complications were 32% of all maternal deaths(17). The current law allows abortion in case of rape, incest, and fetal impairment; if a woman has physical or mental disabilities; or if she is younger than 18 years old. It is the policy that semi liberalizes abortion(17)(18). After this, the number of abortions increased by some amount especially among the age group of 15-19. From other pieces of evidence, one in ten youth reported abortion concerning different socio-demographic factors(19). In 2014, the number of adolescents seeking post-abortion care due to clandestine abortion was greater than those aborted legally (50% vs 18%); between learned and unlearned (14% vs 5%). Data in the same year showed, all types of abortion of all ages increased from 27% to 53%, and was 20% for adolescents. And we observed 96,000 induced abortions and 35,000 clandestine abortions(20). However, the fact that most Ethiopians are religious and have culturally strong identities, we might saw clandestine abortion with complications, and the real could be very large(4). Thus, considering these backgrounds, more work remained ahead and studies like this might be helpful to increase information related to these gaps for future policymaking decisions so that adequate youths might obtain the service.

(4) Language: there is a lot of 'youth' or 'teen' used in this article- even where it is not applicable or appropriate (e.g., ages 15-24. what is understood by 'teen' here?). Youth as a 'catch all' term obfuscates- especially if the data are specific to young women and girls' experiences? If so, that needs to be accounted for clearly in the text- and the gendered implications of this

We mistakenly used the term ‘teen’ and ‘youth’ interchangeably in some of the contexts. After, it was brought to our attention by esteemed reviewer #2; we analyzed and agreed on the term ‘youth’ instead and changed all of it in the area where it was not appropriate.

Thank you once again for all the critical comments and your time. We appreciate your expertise work and professionalism of both reviewers and grateful for that.

Authors,

---

## [Decision Letter · Decision Letter 1]

16 Feb 2021

PONE-D-20-40278R1

Determinants of abortion among youth 15-24 in Ethiopia: A multilevel analysis based on EDHS 2016

PLOS ONE

Dear Dr. Kaso,

Thank you for submitting your manuscript to PLOS ONE. After careful consideration, we feel that it has merit but does not fully meet PLOS ONE’s publication criteria as it currently stands. Therefore, we invite you to submit a revised version of the manuscript that addresses the points raised during the review process.

There is still interest in your study. However, there are revisions that remain to be made. Notably, the authors need to contact a copyeditor that will review the English grammar and syntax. Failure to do so will prohibit acceptance of this article. Please address all of the reviewers concerns in your revised manuscript.

We look forward to receiving your revised manuscript.

Kind regards,

Frank T. Spradley

Academic Editor

PLOS ONE

Reviewers' comments:

Reviewer's Responses to Questions

**Comments to the Author**

1. If the authors have adequately addressed your comments raised in a previous round of review and you feel that this manuscript is now acceptable for publication, you may indicate that here to bypass the “Comments to the Author” section, enter your conflict of interest statement in the “Confidential to Editor” section, and submit your "Accept" recommendation.

Reviewer #1: All comments have been addressed

2. Is the manuscript technically sound, and do the data support the conclusions?

Reviewer #1: Yes

3. Has the statistical analysis been performed appropriately and rigorously? 

Reviewer #1: Yes

4. Have the authors made all data underlying the findings in their manuscript fully available?

Reviewer #1: Yes

5. Is the manuscript presented in an intelligible fashion and written in standard English?

Reviewer #1: Yes

6. Review Comments to the Author

Reviewer #1: Any comment had been tirelessly attempted by the writers. And it was decent but still a significant revision of the grammar, and the discussion part is required

7. PLOS authors have the option to publish the peer review history of their article (what does this mean?). If published, this will include your full peer review and any attached files.

Reviewer #1: **Yes: **Zewdu Girma Shifaw

---

## [Author Response · Author response to Decision Letter 1]

20 Feb 2021

Abstract 

Determinants of the magnitude of abortion among women of diverse social and economic status, particularly in Africa poorly understood because of the missing information in most countries. In this study, we addressed abortion and its determinants among youth women of 15-24 ages to provide clear direction for policymaking in Ethiopia.

Methods: We examined the 2016 Ethiopian demographic health survey data downloaded from the EDHS website after obtaining permission on abortion among 15-24 age women. We applied bivariate and multilevel binary logistic regression. Community and Individual level abortion predictors passed through a three-level binary logistic regression analysis where we used p-value <0.05 and adjusted odds ratios (AOR) with 95% confidence intervals (CI).

Result: The abortion among the youth population in this study was 2.5%. Factors associated with pregnancy were age group 20-24 2.5(1.6-3.8), youth with one birth 0.65(0.44-0.96), youth with 2-5 births 0.31(0.18-0.55), age ≥18 0.50(0.33-0.76), married 38(17-84), divorced 20(7-55), birth in the last five years 0.65(0.44-0.96), middle wealth youth 1.7(1.0.4-2.8), being in Amhara0.31(0.11-0.85), and 0.30(0.12-0.77).

Conclusion: Less abortion occurred in economically poor youths. It is a noble finding; however, the access problem might lead to the result. We observed more abortions in age <18years; those have not given birth until the data collection date. It portrays forth clear policy direction for politicians and all other stakeholders to intervene in the problem. The analysis also showed abortion increased with age. It shows that as age increased, youths disclose abortion which is rare at an early age, and again given an essential clue for the next interventions. The fact in this study is both age and marriage affected abortion similarly. It might be because of various culture-related perceptions where it is not appropriate for an unmarried woman to appear with any pregnancy outcome as the reason behind the decreased number of abortions at a younger age. Thus, more attention is required during implementation for unmarried and lower age youth regardless of the magnitude of the abortion.

. 

 Keywords: pregnancy; abortion; youth; Ethiopia

Introduction

Abortion is an expulsion of pregnancy tissue and products of conception from the uterus intentional by acts spontaneously[1]. Worldwide, abortion is the proven cause of maternal death[2]. Abortion procedures can be safe or unsafe and studies associated every type with the risk of youth mother and fetus deaths. Previously shreds of evidence showed that pregnancies from youth subjected to frequent termination because of physically and psychologically incomplete growth and development[3]. The problem associated with abortion increase and access is to service is subjective laws and cultural aspects of the communities across the world[4]. Legalizing abortion showed improvement in access and usage but did not reduce the number of estimated abortions[5].

 Abortion has different outcomes on the future of youth mothers and the life of the fetus. Other than termination, conceiving during the youth age is also higher risk and correlated with adverse pregnancy outcomes [6]. Some studies associate youth pregnancies with hypertension, eclampsia, premature onset of labor, fetal deaths, premature birth, and pre-eclamptic toxemia delivery outcomes[7]. 

Individual and community-level factors reported responsible for engagement in abortion and pregnancy terminations among which socio-demographic status, resources, early age marriage, and sexual intercourse dominated[8]

. Family background, age, social environment, emotional wellbeing, educational status, employment status, and relationships were the variables reported significantly associated with early age youth pregnancy and abortion[9-12]. A study in Gahanna showed education, religious beliefs, health, economic factors, and family factors are responsible. Other factors also determine the decision to undergo abortion [13]. In other studies, knowledge of abortion rights and services increase usage, although quality, accessibility, and safeties of the abortion also affect the decision to pick the service[14]. In Sub-Saharan countries, women who can make reproductive health decisions are more likely to undergo abortion services[15]. 

Before 2005, the Ethiopian government legalized only the procedure of aborting, in case it required to save the life of a woman[16]. During that time, abortion-related complications were 32% of all maternal deaths[17]. The current law allows abortion in case of rape, incest, and fetal body impairment and if a woman has a physical or mental disability; or if she is younger than 18 years old. It is the policy that semi-liberalizes abortion[17-18]. The number of abortions increased by a substantial amount among the age group 15-19years following the new law. One in ten youth reported abortion concerning different socio-demographic factors[19]. In 2014, the number of adolescents seeking post-abortion care because of clandestine abortion was higher than among those aborted legally (50% for 18%); between learned and unlearned (14% for 5%). Data in the same year indicated many abortions of all ages increased from 27% to 53% and was 20% for adolescents. And we observed 96,000 induced abortions and 35,000 clandestine abortions[20]. However, most Ethiopians are religious and have a culturally strong identity, that clandestine remained unknown. Abortion with complications may only appear that the reality is very different. [4]. Thus, considering these backgrounds, more work remained ahead. Studies like this might help to increase information related to these gaps for future policymaking decisions so that adequate youths might get the service. Therefore, determining abortion among the youth and its associated variables expected to be a great asset for this state of affairs in Ethiopia.

Methods and materials

Design 

We used a cross-sectional study design based on the data from Ethiopian demographic health survey (EDHS) 2016

Setting and participants 

 The Ethiopian population is 112.0 million in 2019 as per the National Bank of Ethiopia and the World Bank. There are nine regions (Tigray, Afar, Amhara, Oromia, Somali, Benishangul, SNNPR (south nation nationalities people’s region), Gambela, and Harari) and two city administrations (Addis Ababa and Dire Dawa) in the country. The administration levels went from regions, zones, and through woredas

[21]. We downloaded EDHS 2016 dataset for this study purpose and extracted youth females of age 15-24 years consider only those who had complete records and then cleaned and made the data ready for the analysis. In this process, we extracted 6,401 population [22]. EDHS collects data on fertility and childhood mortality levels, fertility preferences, awareness, approval, use of family planning methods, maternal and child health, domestic violence, knowledge, and attitudes toward HIV/AIDS and other sexually among the adult population. The frame of the Population and Housing Census (PHC) contains a complete list including information about the enumeration area (EA) location, type of residence (urban or rural), and the estimated number of residential households which developed for this purpose by the Central Statistical Agency (CSA) used[22]. We accessed the data from the Demographic and Health Survey (DHS) website. It is available at (http://www.dhsprogram.com) requesting registration for permission. The data we got then used only for the research purpose. We kept all data confidential, and we did not identify households or individuals. EDHS approved by the Ethiopian Health Nutrition and Research Institute (EHNRI) Review Board and the National Research Ethics Review Committee (NRERC) at the Ministry of Science and Technology, Ethiopia. As published in the survey report of 2016, they collected verbal informed consent from participants, and the purpose of the study was clear to participants [22]. Participation in the survey was voluntary, and they respected the right to decline.

Study variables and data management

The outcome variable for this study was abortion. We took it from the EDHS question ‘have you ever had a pregnancy termination?’ with the response ‘yes’ if the woman ever had an abortion and otherwise ‘no’ as a binary outcome. It usually means abortion is any pregnancy outcome of a miscarriage, abortion, or stillbirth[23-25]. The exploratory variables are individual or group variables showing both mother and child, every socio-demographic variable used in the selected EDHS dataset. After downloading the dataset and including it in the study according to the criteria, we cleaned data in Stata v. 15.0. The data then weighted as per sampling weight, primary sampling unit, and strata before analyzing in Stata 15.0 

Finally, we examined abortion in 2016 datasets and discovered the correlation of independent with outcome variables. Individual and group-level predictors of abortion examined using multilevel logistic regression on pooled data from the datasets. Significance level maintained at p<0.05 with 95% confidence intervals (CI). Before the multilevel logistic regression application, we checked all assumptions. Each variable checked on bivariate before introducing into the consecutive multilevel logistic regression models where 0.2 used to include variables to models. 

Results

We pooled 6,401 female populations from EDHS 2016 dataset picked for the current study to examine abortion status in age 15-24. The proportion of abortions became 2.5%. Community-level weighted frequency analyses showed a higher number of participants were from Oromo (36.3%), Amhara (22.5%), and SNNP (20.37%) regions. Three-fourth of the participants were from urban residences. At the individual level, the age group of 15-19 years was 55% of all participants. Those who learned primary education (54.25%); those with no education also accounted for 20%. More than half (56%) of them included in this study were single; 39% were married. Eighty-four percent of youth started sexual intercourse earlier than 18years; 72% had no work during the survey; 48% were in the rich wealth status category; 32% in unfortunate economic status. Among the youth who participated in the study, around 70% have never given birth; (Table1). 

Multilevel analysis 

The null model had 15.7% of the total abortion variations resulted from the variabilities among clusters and the rest from individual differences. The clustering effect shown up here directed us to take multilevel analyses. The median odds ratio of abortion in the null model was 2.1 affirming the differences among the clusters. We cannot be sure if the results of two randomly picked different or similar as we cannot rule out the between cluster variations.

We followed four model building steps to account for the inter-cluster variation stated above. The null model is an intercept-only model that has given the clue of continuing model development. In model one or the individual level, females in the age group 20-24 years were 2.5 times more likely to have an abortion with an AOR of 2.5 (1.6-3.8). Considering birth in the last five years, female who gave at least one birth showed 35% reduced abortion with an AOR of 0.65 (0.44-0.96), and those who gave to 2-5 birth had 69% reduced abortion with an AOR of 0.31 (0.18-0.55) compared to no birth category. Similarly, the odds of being in the abortion group was 45% reduced for age ≥18 years with an AOR of 0.55 (0.36-0.82) than those <18. In contrast, youths who married had 32 times more likely to have an abortion with an AOR of 32(14-70), and those who divorce were 16 times more likely to experience abortion with an AOR of 16.6(6-44) compared to single.

And in model two that is a community level, Amhara, Benishangul, SNNP, and the Gambella had 66%, 71%, & 54% reduced abortion with AOR of 0.34(0.15-0.75), 0.29 (0.11-0.75), 0.46(0.23-0.94), and 0.28 (0.10-77) respectively compared to the Somali region.

In the ultimate or mixed model, females in the age group 20-24 were 2.5 times more likely to have an abortion with an AOR of 2.5(1.6-3.8) relative to age 15-19. The odds of abortion among wealthy youth were 1.7 times more than the odds of poor wealth status females with an AOR of 1.7(1.0.4-2.8). Considering birth in the last five years, females who gave at least one birth had 35% reduced abortion with an AOR of 0.65(0.44-0.96); those who gave 2-5 births had 69% reduced abortion with an AOR of 0.31(0.18-0.55) compared to no birth category. 

The odds of being in the abortion group were 50% reduced for age ≥18 years old with an AOR of 0.50(0.33-0.76) compared to age <18. And as in model 1, youths who married had 38 times more likely to report abortion with an AOR of 38(17-84); those who divorced were 20 times more likely to have an abortion with an AOR of 20(7-55) compared to single marital status. And for community level, youth females in the Amhara region were 69% reduced abortion with an AOR of 0.31(0.11-0.85); those living in Benishangul had 70% a reduced abortion with an AOR of 0.30(0.12-0.77); (Table2)

In table 3 below, we showed the comparison of each effect of model 0-3. We saw decreased variance from ICC, media odds ratio, and deviance. The log-likelihood ratio and proportional change in variances increased as global expectation. Specifically, the lower deviance shows good model fitness.

Discussion

 Our study examined abortion both at the individual and community levels. For this purpose, we pooled 6,401 samples of age 15-24years female youth population from the Ethiopian demographic health surveillance dataset of 2016. Participants of this study are within the age group of 15-24 years old; however, 40% of them were in marriage and exposed to different youth pregnancy-related problems like abortions at most. The analysis showed, 2.5% of them were at least aborted once. The magnitude is incongruent with the studies conducted in Ethiopia and Nigeria[23,26,27]. The reason might be the socio-cultural influences where some youth might follow clandestine abortion in some areas. Economically, 33% of youth stayed poor; 73% were workless during the survey. It is consistent with a study by Van Rensburg that showed societal poverty, unemployment, and other socio-demographic factors affected youth pregnancies and might lead to abortion[28]; (Table1) 

Abortion was increased with increased age so that age group 20-24 had more abortion than15-19 The finding is in agreement with that of Nigerian DHS where abortion was 2.34 higher among those aged 20–24 than youths aged 15–19 [23]. It might show that age is the risk of abortion during the younger time because of the lack of information and less self-relying decision making.

 We found middle wealthier youths experienced more abortions compared to the poor ones. Other Studies suggested that abortion has been affected by economic status. It was not in some analyses of the same dataset [19-21]. It means youths with financial capacity could abort in case of induction than those who are poor. Youth mothers who gave at least one or more births in the last five years had more abortion history. Studies in Mozambique and Ghana agree with this finding [31]. The reason might be the increased frequency of pregnancy increased the risk of abortions in married youths. Much of the association was among the age group of greater than 18 years [22, 23]. These might be due to the increase in knowledge and self-responsibilities when youth are more mature. Similarly, both the married and the divorced youth mothers had an increased risk of abortion relative to the unmarried ones. The finding is consistent with other studies in Ethiopia, another Africa, and non-African countries (22,23,33]; (Table2)

The fact that divorce youth had an increased abortion might indicate freedom of decision-making on reproductive matters. We compared other regions to Somali so that Amhara and Gambella showed less abortion correlation. From 2000-2016 EDHS pregnancy termination was lower in these regions(Amhara & Gambella)[19]. The event was not so clear whether it is due to actual number or cultural restriction more strong in these regions. However, the information obtained from this study might be additional paramount for further investigation and policymaking. EDHS follow a kind of data collection and methodologies which are internationally standard and scientifically applied in every country. The clustering effect might be another issue during sampling. Furthermore, the economic status classification was country-specific. However, to account for the limitations, we applied weighting technique and Segregated or multilevel analysis to handle variations among clusters; (Table3) 

Conclusion 

 . We provided information about community and individual/ contextual level analyses of abortion among women aged 15–24 years in Ethiopia. The findings were relatively lower than other African countries like Nigeria, Uganda, and South Africa; but, it is necessary to remember that we used weighted data for analysis and reveals a much more country-level profile. At an individual level, we recognized association with age, wealth index, marital status, birth in the last five years, and age at first sexual intercourse. Amhara and Benishangul region have shown an association with youth abortion. The analysis also showed abortion increased with age. As age advances, youth disclose abortion increases that an essential clue for the next interventions related to youth economic status. Various cultures influenced perceptions and inability to make reproductive decisions and need more attention behind the decreased abortion of lower age and unmarried youth. The authors look forward to large-scale studies that consider abortion disaggregation within primary data to account for the current limitations.

Acknowledgments

The authors are very thankful for the responsible DHS for they have given permission to use the data and all other stakeholders involved directly or indirectly.

Funding

We received no funds for this work.

Availability of data and materials

The data used in this study are third-party data from the Demographic and Health Services (http://www.dhsprogram.com) and accessed by following the protocol outlined in the Methods section.

Authors’ contributions

GG developed the proposal, writing results, and drafting the manuscript while SHM was involved in the conception, analysis, and revising of the final manuscript.

Consent for publication

Not applicable

Competing interests

The authors declare that they have no competing interests.

---

## [Editor Report · Decision Letter 2]

23 Feb 2021

Determinants of abortion among youth 15-24 in Ethiopia: A multilevel analysis based on EDHS 2016

PONE-D-20-40278R2

Dear Dr. Kaso,

We’re pleased to inform you that your manuscript has been judged scientifically suitable for publication and will be formally accepted for publication once it meets all outstanding technical requirements.

Kind regards,

Frank T. Spradley

Academic Editor

PLOS ONE

---

## [Editor Report · Acceptance letter]

26 Feb 2021

PONE-D-20-40278R2 

*Determinants of abortion among youth 15-24 in Ethiopia*: A multilevel analysis based on EDHS 2016 

Dear Dr. Gilano:

I'm pleased to inform you that your manuscript has been deemed suitable for publication in PLOS ONE. Congratulations! Your manuscript is now with our production department. 

Kind regards, 

on behalf of

Dr. Frank T. Spradley 

Academic Editor

PLOS ONE